# FLAIR: A Foundation Model for Grapheme Recognition in Ancient Scripts with Few-Shot Learning

## Abstract

The Indus Valley Civilization (IVC) left behind an undeciphered script, posing a significant challenge to archaeologists and linguists. This paper introduces FLAIR, a few-shot learning approach that aims to establish a foundational model for recognizing and identifying individual graphemes from the limited available Indus script. As a foundational model, FLAIR is designed to be versatile, supporting multiple potential applications in script recognition and beyond. It leverages prototypical networks combined with a modified proposed encoder network for segmentation, *ProtoSegment* to extract intricate features from the grapheme images. We evaluate FLAIR's ability to generalize from minimal data using IVC grapheme classification tasks and further experiment with pre-trained Omniglot models for fine-tuning. Additionally, we simulate real-world data scarcity by intentionally restricting training data on the Omniglot dataset. Our experiments demonstrate FLAIR's accuracy in digitizing and recognizing Indus Valley seal graphemes, outperforming traditional machine learning classification approaches. These results underscore FLAIR's potential not only for the digitization of ancient scripts with limited labeled datasets but also for broader applications where data is scarce. FLAIR's success in grapheme recognition highlights its promise as a foundational model capable of extending to other undeciphered writing systems, thereby contributing to the integration of classic scientific tools and data-driven approaches.

## 1 Introduction

The history and civilizations of the past are preserved mainly in the languages of the past. But their field is laborious, requiring specialists to work on a variety of demanding text-based tasks, such as determining the authors of literary works, restoring damaged inscriptions, and translating lost languages Sommerschield et al. (2023). The texts that remain preserved to this day were written in a variety of scripts (Brahmi, Old Chinese, Egyptian hieroglyphs, ancient Greek, Indus, Latin, Mayan, and others) and on a range of materials (bone, metal, palm leaf, paper, papyri, parchment, potsherds, stone, etc.). Technological innovation in machine learning has brought about revolutionary breakthroughs in the study of ancient languages and texts over the past 20 years. Modern Handwritten Text Recognition (HTR) methods struggle to recognize manuscripts with uncommon scripts or alphabets Sánchez et al. (2014); Bhunia et al. (2021); Souibgui et al. (2022); Kang et al. (2020). Conventional machine learning techniques rely on vast quantities of labeled data for training which presents a significant challenge for the Indus script due the scarcity of useful data. Generative AI, such as GANs or diffusion models, can synthesize training data when real examples are scarce. However, generating synthetic data for ancient scripts like the Indus script is challenging due to the lack of deep knowledge of the script's visual and contextual nuances, which remain speculative. Without a substantial labeled corpus, generative models trained on limited data may produce unrealistic or misleading samples that fail to capture the script's true variability. Additionally, generative models require extensive training on diverse examples to generate high-quality outputs, a need that the limited dataset of the Indus script cannot meet. This can lead to artifacts that distort model performance rather than enhance it. Few-Shot Learning (FSL) provides a compelling solution to overcome this challenge. Unlike traditional models, FSL excels in precisely the situation we face – limited data.

These models are specifically designed to learn complex patterns from a remarkably small number of labeled examples per class. There's a possibility of encountering previously unseen symbols during the digitization process. The FSL model can potentially adapt and classify these new symbols based on its learned knowledge from similar classes.

The Indus Valley seals contain intricate patterns of graphemes and motifs, where the script remains largely undeciphered, hindering our understanding of one of the world's oldest civilizations Oakes (2017); Daggumati & Revesz (2021). Unfortunately, despite sustained efforts from archaeologists and linguists, the Indus script remains stubbornly undeciphered Varun Venkatesh & Ali Farghaly (2023).The limited corpus of inscriptions, coupled with the absence of a bilingual Rosetta Stone equivalent, has compelled researchers to explore alternative approaches, such as statistical analyses of grapheme sequences, intra-script grapheme associations, and contextual clues derived from archaeological artifacts Rao et al. (2009; 2010; 2015). These manual efforts, while insightful, are labor-intensive, time-consuming, and limited in scalability. Furthermore, the existing collection of Indus Valley texts is frustratingly limited.

In this context, FLAIR is introduced as a foundational model for ancient script recognition, addressing a significant gap in the field. To the best of our knowledge, there is no widely recognized foundational model specifically tailored for OCR or grapheme recognition that matches the versatility and adaptability seen in foundation models from other domains, such as NLP or general image processing. While existing few-shot learning (FSL) architectures, like Prototypical Networks Snell et al. (2017), are designed to classify new instances based on their similarity to learned prototypes, they may struggle to capture the intricate features and complexities of ancient script characters, particularly when training data is limited. To address these limitations, we introduce *ProtoSegment*, a novel few-shot learning approach that enhances prototypical networks with a segmentation encoder. This modification enables the model to extract intricate features from graphemes (individual characters) in the Indus Valley script, leading to improved identification. By incorporating a segmentation encoder, ProtoSegment can better capture the subtle details and variations within each character class, even with limited training data. The segmentation encoder in ProtoSegment is designed to identify and segment individual graphemes within the script. This segmentation process allows the model to focus on the relevant visual features of each character, improving its ability to distinguish between different classes. To evaluate the effectiveness of ProtoSegment, we conduct experiments on two datasets. We have utilized the Omniglot dataset, a rich collection of handwritten characters and mirrored the real-world data constraints of the IVC script by intentionally restricting it. This controlled setting allows us to assess the model's ability to generalize effectively with minimal data, simulating the IVC grapheme recognition task. Our results demonstrate that ProtoSegment outperforms existing few-shot learning and deep learning methods on both datasets, achieving higher accuracy in grapheme classification tasks.

## 2 Related Work

**Few-Shot Learning with Limited Data:** Recent years have seen various methods developed for learning deep networks with scarce data. Taigman et al. Taigman et al. (2014) and Koch et al. Koch et al. (2015) approached this as a verification problem, using Siamese neural networks Bromley et al. (1993) to determine whether two samples belong to the same class by measuring the distance between them in the learned embedding space. Huang et al. Huang et al. (2019) introduced Deep Prototypical Networks (DPN) to address data insufficiency and class imbalance by capturing discrepancies across classes in a main embedding space. DPN was further enhanced with a masking module for robust classification, though it does not yet incorporate external knowledge sources. Researchers Pahde et al. (2021) have also designed a cross-modal feature generation framework that enriches low-population embedding spaces in few-shot scenarios by mapping text data into the visual feature space using generative models. Ji et al. Ji et al. (2020) proposed Improved Prototypical Networks (IPN), incorporating an attention-based strategy to better capture intra-class distribution by assigning weights to samples based on their representativeness.

**Data-Level Approaches:** A more logical approach to few-shot learning is to use a data-level approach, which means that by gathering more relevant data, the model's performance can potentially be enhanced. In addition to the initial training set, Douze et al. Douze et al. (2018) developed a semi-supervised strategy that incorporated a sizable unlabeled dataset of comparable images. This

vast data collection was used in the few-shot learning scenario to facilitate label propagation. By creating the squared gradient magnitude loss, which drives models to generalize successfully from only a few samples, Hariharan et al. Hariharan & Girshick (2017) merged both strategies (data-level and algorithm-level) and, created new images by hallucinating features. In order to provide new training data for the latter, they trained a model to identify common transformations between preexisting images.

**Meta-Learning Techniques:** Meta-learning techniques have been used in other contemporary few-shot learning methodologies. In a few-shot learning environment, a long short-term memory (LSTM) network was trained as a meta-learner Ravi & Larochelle (2016) to learn the precise optimization technique for training a learner neural network that carries out the classification. The discovery that the update function of common optimization techniques, such as SGD, is comparable to the updating of an LSTM's cell state led to the proposal of this technique. Finn et al. Finn et al. (2017) proposed a model-agnostic meta-learning technique (MAML) that trains a model on base classes and then refines it on a limited number of unique classes to achieve optimal performance. Furthermore, using a few-shot learning technique, Bertinetto et al. Bertinetto et al. (2016) trained a meta-learner feed-forward neural network to predict the parameters of another discriminative feed-forward neural network.

**Attention Mechanisms:** Another technique that has been applied successfully to few-shot learning recently is attention Wang et al. (2022); Vaswani et al. (2017). To identify prototypes, Arık and Pfister Arik & Pfister (2020) use an attention mechanism that compares the encoded representations to samples. By adding a relational attention mechanism to an encoder, prototypical learning enables novel capabilities. Sparsemax attention increases robustness to label noise and allows for the basis of learning on a small number of relevant samples that may be returned at inference for interpretability.

**Prototypical Learning**: The principle of ProtoSegment is inspired by (Badrinarayanan et al. (2017); Feng et al. (2021); Chang et al. (2020)) where they emphasize discarding the fully connected layers in favour of retaining higher resolution feature maps at the deepest encoder output. Similar modifications have been made for prototypical networks based on varying application domain spaces Arik & Pfister (2020); Ji et al. (2020); Tang et al. (2023); Ke et al. (2021); Du et al. (2023).

**Ancient Script Recognition Approaches**: There have been a few approaches developed, with reference to few-shot learning approaches for Ancient Script Recognition. Hu et al. Wenbo Hu et al. (2023) proposed a Visually Guided Text Spotting (VGTS) approach that accurately spots novel characters using just one annotated support sample. Souibgui et al. Mohamed Ali Souibgui et al. (2020) use few-shot object detection for the task of handwritten ciphers recognition. The method includes detection of all symbols of a given alphabet in a line image, and then a decoding step to map the symbol similarity scores to the final sequence of transcribed symbols. They use the Omniglot dataset Yang Li et al. (2021) to create synthetic query lines that simulate handwritten ciphered lines. The study by Varun Venkatesh et al. Venkatesh & Farghaly (2023) investigated the Indus script by analyzing patterns and positions of individual signs, pairs, and sequences. They built statistical models and algorithms to predict sign behavior based on their position. This analysis revealed significant differences in the language used in Indus texts from West Asia compared to those from the Indian subcontinent, suggesting distinct regional dialects within the Indus civilization. Ansari et al. Ansari et al. while being not directly related to deep learning, provides comparative visual analysis with valuable insights for future deep learning approaches. By comparing the visual features of Indus symbols with those from other writing systems, researchers Rao et al. (2009) had bearings on identifying potential similarities in form or structure. This comparative analysis can inform the design of deep learning models by highlighting specific visual characteristics that the model might focus on when analyzing Indus script characters. Palaniappan & Adhikari (2017) Palaniappan & Adhikari (2017) address the time-consuming task of creating standardized corpora for undeciphered scripts like the Indus Valley Script. They propose a deep learning pipeline to automate this process. The pipeline segments images into regions, classifies them as textual or not, refines textual regions, isolates individual symbols, and classifies them based on a reference corpus. While achieving 92% accuracy for identifying a specific symbol, this work demonstrates the initial potential of deep learning to expedite corpus creation and advance research on the Indus Valley Script.

**Our contribution:** In contrast to prior work that primarily focused on character detection or corpus creation, FLAIR directly addresses the core task of grapheme recognition. The integration of a segmentation encoder within the prototypical network architecture enables the model to capture

finer-grained features and spatial relationships within graphemes, leading to improved recognition performance even with limited labeled data. The development and evaluation of FLAIR on the IVC dataset establishes a benchmark for future research in this domain and contributes to the advancement of efforts to decode the Indus script and other undeciphered writing systems.

# 3 METHOD

We present the methodology for IVC Script grapheme recognition in Figure 1. Our approach leverages few-shot learning to address the challenge of limited labeled data in this domain. We begin by curating a dataset of Indus Valley script graphemes, drawing from Parpola's CISI volumes Joshi et al. (1987) and Mahadevan's seminal work, "The Indus Script: Texts, Concordance and Tables" Mahadevan (1977). This dataset comprises 262 images distributed across 39 classes, each meticulously annotated to delineate individual graphemes. These annotations, stored in XML format, enable an automated script to crop and classify each grapheme into one of 39 distinct classes as defined by Mahadevan (Figures 2, 3).

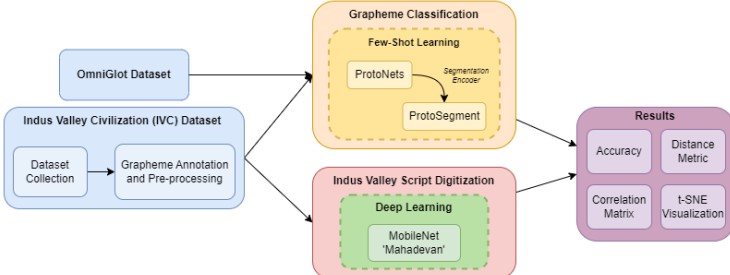

Figure 1: IVC Script Grapheme Recognition and Methodology

To facilitate the digitization of this dataset, Atturu Atturu (2024) developed ASR-Net for grapheme identification, which is based on *MobileNet*Sinha & El-Sharkawy (2019). These tools automate the digitization of Indus seals, providing researchers with efficient means to analyze vast collections of artifacts and glean insights into the socio-cultural and economic facets of the Indus Civilization. Complementing these efforts, a comprehensive database of high-resolution Indus seal images has been established, complete with metadata detailing provenance, dimensions, and associated inscriptions. This database serves as a cornerstone for Indus Valley research, offering a rich repository of visual and contextual data for training and validating our machine learning models.

For few-shot learning on this IVC dataset, we employ two models: re-implemented *ProtoNets* Feng et al. (2021) from the literature and our proposed *ProtoSegment*, a novel extension of *ProtoNets*, which incorporates a segmentation encoder network for enhanced feature extraction. Both models are trained to learn prototypical representations of each grapheme class from limited labeled data. During inference, new images are classified by comparing them to these learned prototypes. This approach aims to achieve state-of-the-art performance in grapheme identification on the challenging data starved IVC dataset. We expand on the individual methodology blocks in further sections.

## 3.1 IVC DATASET PRE-PROCESSING

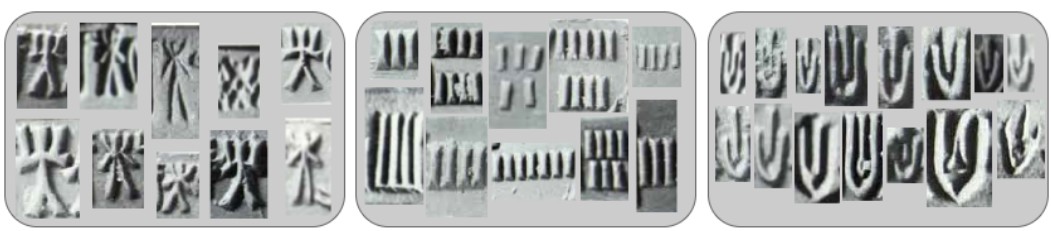

Figure 2: Sample grapheme images for class label *M8, M104, and M336*.

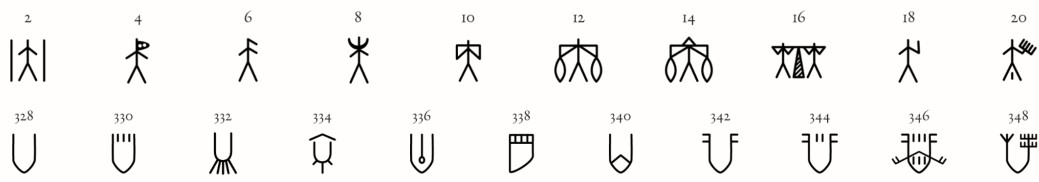

Figure 3: Sample grapheme labels as assigned by Mahadevan Mahadevan (1977)Mahadevan & Research Library

.

The initial dataset of 262 images underwent preprocessing to remove duplicates and annotate individual graphemes within each image. Annotations were stored in XML files, specifying the location and class of each grapheme. An automated script then cropped and sorted the graphemes into 39 classes based on Mahadevan's classification Mahadevan (1977). These classes were selected based on having at least six image samples (3 for query and 3 for support) per class, ensuring sufficient data for training, validation, and testing. The 39 selected class labels are: M8, M12, M15, M17, M19, M28, M48, M51, M53, M59, M102, M104, M141, M162, M173, M174, M176, M204, M205, M211, M216, M245, M249, M267, M287, M294, M296, M302, M307, M326, M327, M328, M330, M336, M342, M387, M389, and M391. The 'M' prefix denotes Mahadevan's classification, followed by the specific identifier assigned by him.

## 3.2 PROTOSEGMENT MODEL

Existing prototype-based neural networks can be architecturally deconstructed into three primary interconnected components. The first component is a convolutional neural network (CNN) $f$, parameterized by a set of weights $w_{conv}$, which serves as a latent feature extractor that processes input images $x$. The CNN $f$ converts each input image $x$ into a set of high-dimensional "patch" vectors $z_i \in R^D$, where each vector $z_i$ corresponds to the latent feature representation of a spatial region or patch from the original input image space. The second core component is the prototype layer $p$, which operates directly on the convolutional output $f(x)$ comprising the set of latent patch vectors $z_i$. Each prototype is the mean vector of the embedded support points belonging to its class k ($c_k$):

$$c_k = \frac{1}{|S_k|} \sum_{(xi,yi) \in S_k} f(\phi(x_i)) \qquad (1)$$

where $c_k$ is the M-dimensional representation, $S = (x_1, y_1), ..., (x_N, y_N)$ is a small support set of $N$ labeled examples and $S_k$ denotes the set of examples labeled with class $k$. The prototype layer compares each patch vector $z_i$ against a learned set of $m$ prototype vectors $P = \{p_j\}_{j=1}^m$, where each prototype vector $p_j \in \mathbb{R}^D$ resides in the same high-dimensional latent space:

$$p\phi(y = k|x) = \frac{exp(-d(f(\phi(x)), c_k))}{\sum_{k'} exp(-d(f(\phi(x)), c_{k_0}))} \qquad (2)$$

as the image patch vectors, where $d$ is the metric function. For every prototype vector $p_j$, the prototype layer calculates a similarity score $g_{p_j} \cdot f(x)$ that is a monotonically decreasing function of the distance between $p_j$ and the closest latent patch vector $\tilde{z} \in f(x)$ in the model's feature space. Learning proceeds by minimizing the negative log-probability:

$$J(\phi) = -\log p\phi(y = k|x) \qquad (3)$$

of the true class $k$ via SGD Snell et al. (2017). Training episodes consist of a subset of classes from the training set that are chosen at random, followed by the selection of a subset of instances from each class to serve as the support set and a subset of the remaining classes to act as query points.

The third component is a prototype class assignment mechanism $h$ that follows the prototype layer $g_p$. This mechanism assigns evidence logits to each output class based on the prototype similarity scores $g_{p_j} \cdot f(x)$ calculated in the previous layer, in conjunction with a set of class assignment weights $w_h$. These evidence logits are then normalized via a softmax function to yield the model's

final predicted probability distribution over the output classes for the given input image $x$. Crucially, in the final instantiation of the model, each prototype vector $p_j$ is constrained to be exactly equal to a specific latent patch vector $\tilde{z} \in f(x_i)$ extracted from the CNN's representation of some training image $x_i$. Specifically, when making a prediction on a test input image $x$, the model is effectively

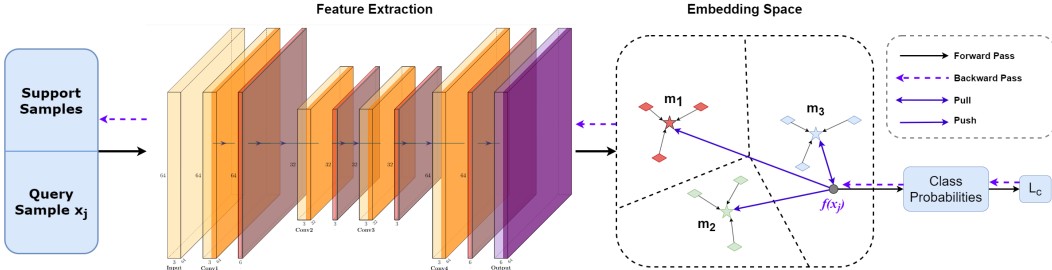

Figure 4: The classification process of ProtoSegment Model

comparing the salient latent features encoded in each patch vector $z_i \in f(x)$ against the salient features underlying the previously seen training image patches that are encapsulated and represented by the prototype vectors $p_j$. The similarity scores $g_{p_j} \cdot f(x)$ calculated by the prototype layer $p$ quantify the degree to which each test patch vector $z_i$ matches or differs from the corresponding prototype $p_j$ in the high-dimensional latent space. Higher scores indicate a closer match between a test patch and a learned prototype. These scores then get converted into class evidence logits by the final assignment mechanism $h$. This overall reasoning process of matching test patches to learned prototypes derived from training examples provides a visually-grounded interpretability mechanism. One can visualize and examine the specific training image patches that each prototype is tied to, in order to understand what high-level semantic concepts or visual patterns that prototype represents and captures.

The architecture of the ProtoSegment model is illustrated in Figure 4. The model consists of a segmentation encoder, a feature extractor, and a prototypical layer. The segmentation encoder $g$ is a crucial component designed to address the challenge of isolating individual graphemes within the intricate Indus script. It employs a convolutional encoder-decoder architecture with skip connections to accurately segment the input image $x$ into distinct regions, each ideally corresponding to a single grapheme. The output of this encoder is a set of segmented regions ($S = s_1, s_2, ..., s_n$), where each $s_i$ represents a distinct grapheme. This segmentation process allows the subsequent feature extractor to focus on individual characters, mitigating the complex nature of the script. The feature extractor, $f$, is implemented as a Convolutional Neural Network (CNN) with four convolutional blocks. Each block consists of a 3x3 convolutional layer, batch normalization, a ReLU activation function, and 2x2 max-pooling. This architecture effectively captures hierarchical features from each segmented grapheme region $s_i$, producing a corresponding 64-dimensional embedding $z_i = f(s_i)$. The choice of 64 dimensions was empirically determined to balance representational capacity and computational efficiency. The prototypical layer, $p$, computes a representative prototype for each grapheme class $k$ by averaging the embeddings of the support set samples belonging to that class:

$$c_k = \frac{1}{|S_k|} \sum_{s_i \in S_k} z_i \tag{4}$$

where $S_k$ is the set of segmented regions belonging to class $k$. The distance between a query sample's embedding $z_q$ and the class prototypes $c_k$ is calculated using a distance metric $d(z_q, c_k)$, such as Euclidean distance. The predicted class for the query sample is then determined by:

$$\hat{y}_q = \arg\min_k d(z_q, c_k) \tag{5}$$

By incorporating the segmentation encoder, ProtoSegment can capture finer-grained features and spatial relationships within graphemes, leading to more distinct embeddings and improved discrimination between visually similar characters.

## 4 EXPERIMENTS AND RESULTS

### 4.1 OMNIGLOT FEW-SHOT CLASSIFICATION

Here we describe how we developed the proposed foundation model by pre-training the prototypical network on the labeled OmniGlot data set. The Omniglot dataset Lake et al. (2011) comprises 1623 handwritten character samples collected across 50 alphabets, with 20 examples per character drawn by distinct human subjects. Following the experimental setup of Vinyals et al. Vinyals et al. (2016), we preprocess the grayscale images by resizing them to $28 \times 28$ pixels and augmenting the character classes through rotations in 90-degree increments. We allocate 1200 characters plus their rotated variants for training (4,800 classes in total), with the remaining classes and their rotations reserved for testing.

Table 1: Few-shot classification accuracies on Omniglot

| Model | Fine Tune | 5-way Acc. | | 20-way Acc. | |
|---|---|---|---|---|---|
| | | 1-shot | 5-shot | 1-shot | 5-shot |
| **MANN Santoro et al. (2016)** | N | 82.8% | 94.9% | - | - |
| **Siamese Nets Koch et al. (2015)** | N | 96.7% | 98.4% | 88.0% | 96.5% |
| **Siamese Nets Koch et al. (2015)** | Y | 97.3% | 98.4% | 88.1% | 97.0% |
| **Matching Networks Vinyals et al. (2016)** | N | 98.1% | 98.1% | **98.1**% | 98.1% |
| **Matching Networks Vinyals et al. (2016)** | Y | 97.9% | 98.7% | 93.5% | 98.7% |
| **Siamese Nets with Memory Kaiser et al. (2017)** | N | 98.4% | 99.6% | 95.0% | 98.6% |
| **Neural Statistician Edwards & Storkey (2016)** | N | 98.1% | 99.5% | 93.2% | 98.1% |
| **Meta Nets Munkhdalai & Yu (2017)** | N | 99.0% | - | 97.0% | - |
| **Prototypical Networks Snell et al. (2017)** | N | 98.8% | **99.7**% | 96.0% | 98.9% |
| **Relation Net Sung et al. (2018)** | N | **99.4**% | 99.7 % | 97.4% | 99.0% |
| **ProtoSegment (Ours)** | N | 98.3% | 99.4% | 95.8% | 98.6% |
| **ProtoSegment (Ours)** | Y | 98.9% | **99.7**% | 96.5% | **99.2**% |

The input block is composed of a $3 \times 3$ convolutional layer with 64 filters, down-sampled to 32 filters and then is up-sampled back to 64 filters, followed by batch normalization Ioffe & Szegedy (2015), a ReLU nonlinearity, and a $2 \times 2$ max-pooling operation. When applied to the $28 \times 28$ Omniglot images, this architecture yields a 64-dimensional embedding space. We utilize the same encoder network for embedding both support and query examples. Model training was performed via stochastic gradient descent with the Adam optimizer Kingma & Ba (2014), using an initial learning rate of $10^{-3}$ that was halved every 2000 episodes. No explicit regularization was employed beyond batch normalization. We trained ProtoSegment Networks under the 1-shot and 5-shot learning scenarios, with each training episode comprising 60 classes and 5 query points per class. We observed improved performance when matching the training-shot to the test-shot, and by using a higher "way" (number of classes) per training episode. For performance evaluation, we computed the classification accuracy averaged over 1000 randomly sampled episodes from the test set. We compared against several baselines, including the neural statistician Edwards & Storkey (2016) and both fine-tuned and non-fine-tuned versions of matching networks Vinyals et al. (2016). The results, presented in Table 1, represent the current state-of-the-art on this dataset to our knowledge.

### 4.2 IVC FEW-SHOT CLASSIFICATION

#### 4.2.1 DEEP LEARNING APPROACH

The initial approach ASR-NetAtturu (2024), employs Convolutional Neural Networks (CNNs) to recognize characters within bounding boxes, leveraging their ability to learn and extract features from images automatically. The MobileNet model is integrated into this architecture to provide further refinement in character recognition. Unlike traditional CNNs that operate on entire images, MobileNet focuses specifically on the characters within bounding boxes, ensuring precise decoding of sequences of graphemes. Additionally, multiple layers of CNN-based classification models are utilized as part of the validation process, working in conjunction with MobileNet to validate and refine the accuracy of character recognition. Furthermore, transfer learning techniques are explored to

enhance the approach's performance. Pre-trained transfer learning-based models, including popular architectures like ResNet and DenseNet, are considered for adaptation and fine-tuning to improve character recognition within bounding boxes. The highest training accuracy is 94% and the highest validation accuracy is 95%. The model has been trained on 40 classes with around 12,264 images with pre-augmentation. The validation data does not undergo the augmentation which has 200 images in total for all the classes. By integrating transfer learning techniques with the ASR-Net model, the initial approach aims to leverage the knowledge and features learned from large datasets, thereby improving the accuracy and efficiency of character recognition in diverse scenarios.

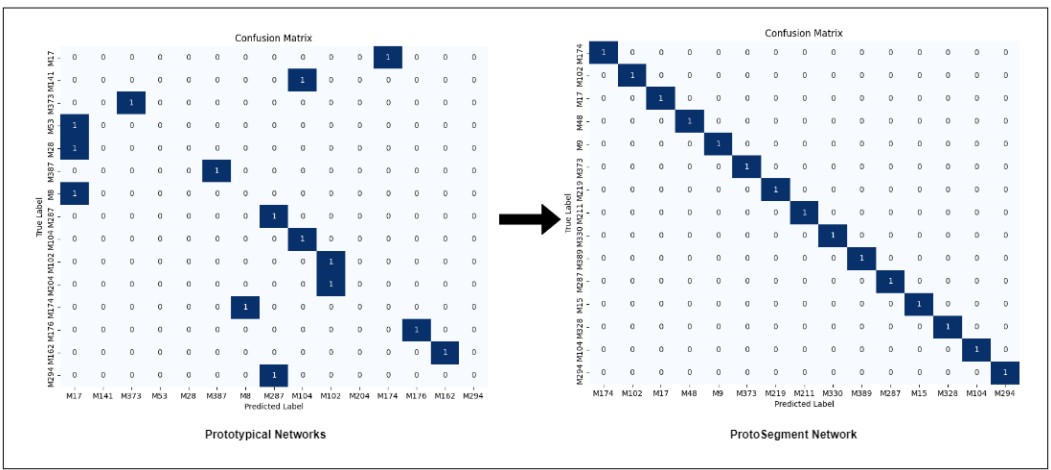

Figure 5: Comparison of confusion matrices for Prototypical Networks and ProtoSegment Network, presented on 15 classes here to save space.

Table 2: Grapheme Classification Accuracies on IVC Dataset. ASR-net is validated only where sufficient data were available.

| Model | 5-way Acc. 1-shot | 15-way Acc. 1-shot | Acc. 12,264 Images |
|---|---|---|---|
| **Prototypical Networks Snell et al. (2017)** | 90.7% | 92.4% | - |
| **ASR-Net DL Atturu (2024)** | - | - | 95% |
| **ProtoSegment (Ours)** | 99.4% | 99.9% | - |

### 4.2.2 FEW-SHOT LEARNING APPROACH

For our few-shot learning approach, we use ProtoSegment and evaluate it against ProtoNets. The model was trained using the Adam optimizer with a learning rate of 0.001 and a gamma value of 0.5. The learning rate was decayed by a factor of 0.1 every 20 steps. The input images were expected to have 3 channels (RGB) with a size of 32x32 pixels and channels were repeated for gray scale images. The model was trained for 200 episodes, with evaluation occurring after every episode. The distance metric used for computing similarities between embeddings and prototypes was the Euclidean distance. During training, the number of classes was set to 5 and 15 (refer to Tab. 2), and the number of query samples per class was set to 1 and the number of support samples per class was set to 1. For validation, the number of classes was also set to 5 and 15 (Tab. 2). The training process was set to run for 100 iterations with a patience of 10 epochs and a minimum improvement threshold of 0.01 for early stopping. The results presented in Table 2, represent the current state-of-the-art on Indus Valley Script to our knowledge for grapheme identification. The testing process involves sampling graphemes classes randomly from the IVC dataset during each iteration. Due to this random sampleing and the limited number of classes being tested per iteration, the resulting confusion matrix (Figure 5) appears near perfect for ProtoSegment. This is because the models are being evaluated on a subset of the entire dataset in each iteration. The confusion matrix indicates

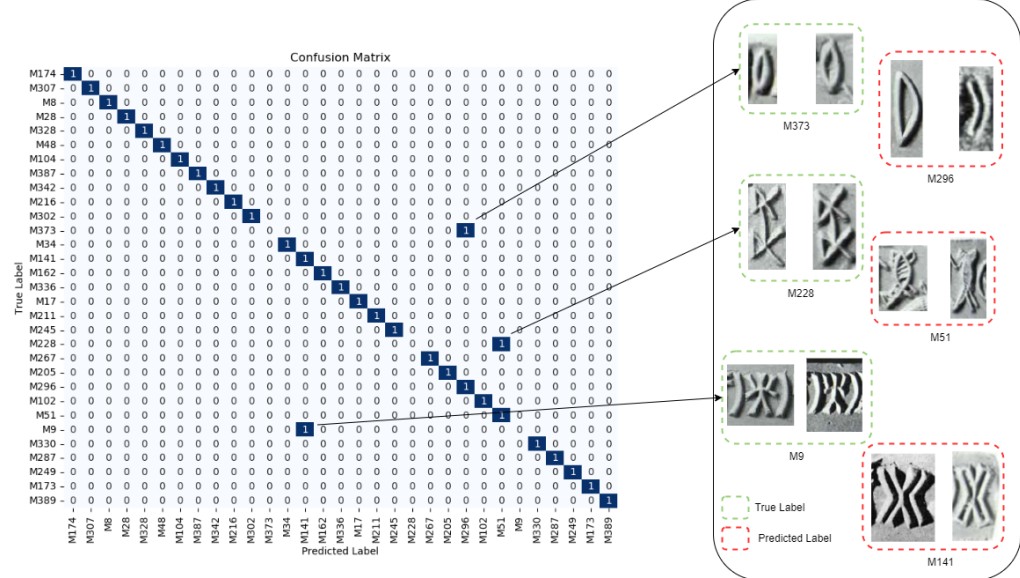

Figure 6: Confusion matrix for Indus Valley Civilization (IVC) grapheme classification for 31 classes, highlighting misclassifications of visually similar symbols (M373, M296, M228, M51, and M9) with corresponding example images from our proposed model. Misrecognized graphemes are visually very similar.

that all grapheme samples within the selected 15 classes for this iteration were correctly classified in the case of ProtoSegment. For each sampled class, one image is used as a query image, and another image from the same class is used as a support image. This process is repeated for 100 iterations and the average accuracy across all iterations is reported as the final performance metric (Table 2). The random sampling of classes in each iteration ensures that the models are evaluated on a diverse set of graphemes, providing a comprehensive assessment of their ability to generalize to unseen samples.

## 5 LIMITATIONS AND SOCIETAL IMPACT

While ProtoSegment demonstrates strong performance in grapheme identification, the confusion matrix (Figure 6) reveals a limitation in distinguishing between visually similar graphemes. For instance, M373 is frequently misclassified as M296, and M228 is often confused with M51. This suggests that the model may struggle to capture subtle differences in stroke patterns or shapes, especially when graphemes share similar overall structures. Addressing this limitation could involve incorporating additional features, such as contextual information or stroke order, to enhance the model's ability to discriminate between visually similar characters. Another potential avenue for improvement lies in exploring alternative segmentation encoder architectures or incorporating attention mechanisms to focus on the most discriminative features of each grapheme. The model's performance is inherently tied to the quality and diversity of the training data. In cases where the available data is limited or biased, the model's ability to generalize to unseen graphemes might be compromised. Additionally, the model's reliance on prototypical representations assumes a degree of visual similarity within each grapheme class. However, variations in handwriting styles and potential degradation of ancient inscriptions could introduce challenges for accurate recognition. From a societal impact perspective, FLAIR's potential to aid in deciphering ancient scripts like the Indus Valley script is significant. By automating and accelerating the process of grapheme identification, FLAIR could contribute to a deeper understanding of ancient civilizations, their languages, and their cultural practices. However, it's crucial to approach the interpretation of deciphered texts with caution, as misinterpretations (e.g., between stylistic variation of a grapheme as a different one) could have unintended consequences for historical narratives and cultural heritage. Upon acceptance of this paper, we will release FLAIR, including the source code, pre-trained model weights, and relevant documentation, on GitHub to ensure transparency and reproducibility of our results.

## 6 CONCLUSION

In conclusion, this paper presents FLAIR as not only a novel approach for Indus script grapheme identification but also as a potential foundational model for OCR. By leveraging few-shot learning and incorporating the modified segmentation encoder network (ProtoSegment), FLAIR demonstrates the capability to achieve state-of-the-art performance even with the limited labeled data available for the Indus script. The model's ability to generalize from minimal examples and its potential adaptability to unseen symbols position it as a powerful tool for not only digitizing and analyzing ancient scripts but also potentially contributing to their decipherment. The development and evaluation of FLAIR on the curated IVC dataset establishes a benchmark for future research, inviting further exploration and refinement. The insights gained from FLAIR's performance can inform the design of future models, potentially incorporating additional features like contextual information or stroke order to further enhance recognition accuracy. By automating and accelerating the process of grapheme identification, FLAIR can significantly contribute to the field of digital humanities, including paleography, epigraphy, and historical linguistics, enabling researchers to efficiently process and analyze large volumes of textual data. This could lead to new discoveries and interpretations of ancient texts, shedding light on the languages, cultures, and histories of past civilizations. While acknowledging the limitations related to data quality and variability, FLAIR's contribution to cultural heritage preservation and its potential for broader applications in deciphering undeciphered writing systems are significant.

ACKNOWLEDGMENTS

This research was partially supported by a grant from the US National Endowment for the Humanities, Award number PR-290075-23.

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
