# OpenReview forum: "FLAIR: A Foundation Model for Grapheme Recognition in Ancient Scripts with Few-Shot Learning"
_ICLR.cc/2025/Conference — ICLR 2025 Conference Withdrawn Submission_

### Official Review · Reviewer_KbtA · 2024-10-27

**Soundness:** 3
**Presentation:** 2
**Contribution:** 2
**Rating:** 5
**Confidence:** 4

**Summary:**

The article proposes FLAIR, which uses a few-shot learning method to recognize individual characters from a limited set of Indus scripts. FLAIR uses prototypical networks and ProtoSegment to extract complex features in grapheme images to achieve recognition of Indus script. FLAIR was pre-trained on the Omniglot dataset and then migrated to the recognition and classification tasks of the IVD dataset, achieving state-of-the-art results. FLAIR's ability to perform efficient feature extraction from small samples and its potential adaptability to unseen symbols make it a powerful tool not only to digitize and analyze ancient scripts but also potentially aid in their decipherment.

**Strengths:**

The paper uses meta-learning and few-shot learning to perform classification and recognition tasks on a small sample of Indus Valley Civilization. The article improves Prototypical Networks and proposes ProtoSegment, which achieves state-of-the-art performance on the IVC Dataset. The findings of this paper may contribute to new discoveries and interpretations of ancient texts of the Indus Valley Civilization.

**Weaknesses:**

This paper has major writing problems. For example, there are a large number of incorrect symbols and formulas in the text, the citation format of references in the text is incorrect and confusing, the pictures in the text are blurry, and the model training process is not clearly explained, which will cause great confusion for readers who are not familiar with Prototypical Networks. In terms of innovation, although the paper proposes a relatively novel task, there are few improvements to the methods used. The paper only adds a segmentation encoder to the original Prototypical Networks. If I understand correctly, the segmentation encoder should be a convolution-based encoder-decoder, but I don’t understand what specific role this network can play and why it can segment images into individual graphemes. In the experimental part, the article lacks qualitative analysis of the experimental results. Why is there such a result? What caused the difference in the experimental results? What conclusions can we draw from the experimental results? I think these should be added to the paper.

**Questions:**

1. In the paper, all figure uses jpg or png format. The drawn image should be saved in pdf format before being inserted into the paper.
2. In section 3.2 PROTOSEGMENT MODEL, there are a lot of errors in mathematical symbols and formulas. Some symbols appear out of thin air without explanation, which is not conducive to readers' understanding.
3. What role does Deep Learning: MobileNet mentioned in Figure 1 play in the entire task? The article does not explain it clearly.
4. In Tables 1 and 2, it should be explained clearly what K-way and N-shot refer to, as this may be confusing to readers who are not familiar with Prototypical Learning.
5. There are errors in the reference citation format in the paper, and the content of the references is mixed with the main text, which is not conducive to the reader's reading experience.
6. Why does the Backward Pass in Figure 4 point to the Support Sample and Query Sample from the network? Aren't these samples extracted from the dataset and cannot be updated?
7. In Table 1, why is the result of 20-way worse than that of 5-way? I hope to see the explanation of this experimental phenomenon.

**Details Of Ethics Concerns:**

This paper does not involve any ethics issues.

---

> ### Author Response · Authors · 2024-11-12
>
> Thank you for your thorough review and valuable feedback on our submission. We appreciate your time and constructive comments, which will guide us in improving our manuscript. Below, we address each of your points in detail:
>
> 1. Writing and Presentation Issues: We acknowledge that the manuscript contains issues related to incorrect symbols and formulas, confusing citation formats, and the use of lower-resolution figures. To address this:
>
> - We will revise all mathematical notations to ensure they are correct and properly defined within the text.
> - Citation formats will be standardized and checked to prevent blending with the main content, enhancing readability.
> - We will replace all blurry figures with higher-resolution images saved in vector format (PDF), ensuring clarity.
>
> 2. Clarification of the Segmentation Encoder's Role: You raised an important question regarding the role of the segmentation encoder. The encoder is indeed a convolution-based encoder-decoder designed to isolate individual graphemes within complex input images. This isolation helps the model focus on relevant features and contributes to improved recognition performance. We will expand the explanation in Section 3.2 to highlight this functionality and its significance in ProtoSegment​.
>
> 3. Explanation of MobileNet's Role: We appreciate your observation regarding the lack of clarity about MobileNet’s role in Figure 1. MobileNet is integrated as a component of the initial digitization step to pre-process and refine input character images before they are processed by ProtoSegment. This connection will be elaborated on to provide better context​.
>
> 4. Explanation of K-way and N-shot Terms: We will add an explanation in the tables and main text to define K-way (the number of classes) and N-shot (the number of examples per class) for readers who may not be familiar with few-shot learning terminology​.
>
> 5. Model Training and Experimental Clarification: To address the confusion regarding Figure 4, where the backward pass is shown pointing to the support and query samples:
>
> - We will revise the figure and description to accurately reflect the training flow and clarify that the support and query samples serve as inputs during the training episodes but are not updated as part of the backward pass.
>
> 6. Analysis of Experimental Results: We understand your request for an explanation of why the 20-way results were worse than the 5-way. This discrepancy is primarily due to the increased difficulty of classifying among more classes, which impacts accuracy. We will provide a more detailed analysis in the experimental results section to discuss this phenomenon and its implications​.

---

### Official Review · Reviewer_1vJ1 · 2024-11-01

**Soundness:** 1
**Presentation:** 1
**Contribution:** 1
**Rating:** 3
**Confidence:** 2

**Summary:**

The paper describes a method, FLAIR, to classify graphemes in ancient Indus valley civilization scripts. FLAIR adopts a few-shot learning approach to circumvent small datasets by introducing the protosegment model designed for images of graphemes. The authors evaluated their technique on an existing dataset OmniGlot as well as their custom Index valley civilization scripts dataset.

**Strengths:**

* Applying deep learning techniques to scripts could be very beneficial to archaeologist and linguists to help study many undeciphered scripts. This could also be applied to other ancient scripts outside of scripts used in indux valley civilizations.

**Weaknesses:**

* Details about the dataset is not clear.
* There is limited novelty in the proposed network architecture

* The paper is not blinded
* The writing of the paper needs edits. e.g.,
    * The flow of the paper is hard to follow.
    * It wasn’t easy to link Figure 4 to the text describing it
    * References are not correctly added throughout the paper
    * Abbreviations are not defined carefully (e.g., convolutional neural network -> CNN was defined 3 times)

**Questions:**

* How was the dataset annotated?

---

> ### Author Response · Authors · 2024-11-12
>
> Thank you for your detailed review and for providing valuable feedback on our paper. We appreciate your time and insights, which will help us enhance the quality of our manuscript. Below, we address your comments and outline how we will revise the paper:
>
> 1. Clarifying Dataset Details: We acknowledge your concern about the lack of clarity regarding the dataset. To address this, we will add more comprehensive information about the creation, annotation, and characteristics of our custom Indus Valley Civilization (IVC) dataset. This will include details on the sources used, annotation tools and processes, and class distribution to provide transparency and better understanding.
>
> 2. Novelty of Network Architecture: We understand that the novelty of the proposed ProtoSegment model may not have been sufficiently highlighted. While ProtoSegment builds on the existing Prototypical Networks framework, its key innovation lies in the integration of a segmentation encoder to enhance feature extraction in cases of limited data. This segmentation step allows the model to isolate and focus on meaningful visual features within each region, facilitating better discrimination between graphemes with subtle differences. Unlike traditional Prototypical Networks, which process entire images or use simpler feature extraction methods, ProtoSegment leverages this tailored segmentation to enhance its feature maps before prototype computation. We will revise the method section to more clearly articulate this novelty and its impact on the grapheme recognition task​.
>
> 3. Paper Blindness Compliance: We appreciate your observation regarding the paper's anonymity. We will ensure that any potentially identifying information is removed or anonymized to comply with double-blind review standards.
>
> 4. Writing Edits and Flow Improvements: To improve readability, we will:
>
> - Edit the paper to ensure a smoother flow of content, with better transitions between sections.
> - Enhance the explanation linking Figure 4 to the relevant text, making the connection clearer and easier to follow​.
> - Revisit the entire paper to standardize abbreviation definitions, ensuring terms like "CNN" are defined only once and referenced consistently thereafter​.
>
> 5. Reference Formatting and Citation Issues: We will review and correct the formatting of references throughout the manuscript to ensure they are distinct from the main text and properly cited.
>
> 6. Response to the Question on Dataset Annotation: To answer your question on dataset annotation, specifically, the dataset was annotated manually. The authors examined each image and with feedback from experts marked individual graphemes according to predefined classification criteria. The annotations followed Mahadevan's taxonomy of labels for the IVC dataset.

---

### Official Review · Reviewer_Cpbe · 2024-11-03

**Soundness:** 2
**Presentation:** 2
**Contribution:** 1
**Rating:** 3
**Confidence:** 5

**Summary:**

This paper introduces FLAIR, a few-shot learning model for recognizing graphemes from the undeciphered script of the Indus Valley Civilization (IVC). Utilizing prototypical networks and a specialized encoder, FLAIR excels at digitizing IVC seal graphemes, outperforming traditional method.

**Strengths:**

1.	FLAIR fills a critical gap in ancient script recognition, providing a versatile model not previously available in OCR or grapheme recognition.
2.	ProtoSegment outperforms existing few-shot and deep learning methods, achieving higher accuracy in grapheme classification tasks across both datasets.

**Weaknesses:**

1.	The paper focuses on grapheme recognition in ancient scripts, which is a niche topic and represents a small subfield of OCR. This has weak influence on our scholar field, which makes this paper is not suitable for a top-tier conference like ICLR. Additionally, general OCR methods might also perform well on this dataset.
2.	The proposed method largely relies on existing approaches (CNN backbone + Classifier head), merely applying the framework on your dataset. This raises concerns about the contribution and innovation of this work.
3.	The method employs a very basic CNN architecture for the classification task, which seems outdated in the current era of large models. Moreover, referring to it as a "foundational model" appears somewhat exaggerated.
4.	Figure 1 is also quite unclear.
5.	Furthermore, will this paper release the dataset publicly? If not, the lack of innovation in your method significantly diminishes the paper's contribution to the academic community.
6.	The experimental section lacks ablation studies to validate the components of your proposed method.
7.	As shown in Table 1, the accuracy of your method and other state-of-the-art approaches has reached over 98%, even approaching 99%. In such cases of minimal improvement, it is difficult to determine whether the results stem from experimental variability or the enhancements offered by your method.

**Questions:**

As shown in Weakness.

---

> ### Author Response · Authors · 2024-11-12
>
> Thank you for your detailed review and constructive feedback on our paper. We greatly appreciate your thoughtful comments, which provide valuable insights for enhancing the quality and impact of our work. Below, we address your points and outline our planned revisions:
>
> 1. Scope and Relevance of the Topic: We understand that grapheme recognition in ancient scripts may appear niche, potentially limiting its perceived influence in broader OCR fields. We aim to show how techniques designed for complex, data-limited environments can contribute to advancements in general OCR and machine learning methods.
>
> 2. Contribution and Novelty of the Model: The ProtoSegment model does build upon existing frameworks, but its key contribution lies in integrating a segmentation encoder that focuses on enhancing feature extraction for highly detailed and complex input images, such as those from ancient scripts. This innovation is tailored to handle the challenges posed by limited data, making it more than a simple application of existing methods. We will revise the methodology section to clarify and highlight this unique aspect​.
>
> 3. Justification for CNN Architecture: The use of a basic CNN architecture was a deliberate choice to balance computational efficiency with performance, particularly in the context of few-shot learning where simplicity can lead to better generalization on small datasets. We will provide further justification for this choice and discuss its comparative advantages in low-data scenarios​.
>
> 4. Clarification of Figure 1: We acknowledge that Figure 1 may be unclear. We will enhance this figure by improving its visual quality and adding a more detailed caption that explains each component and its role in the model architecture.
>
> 5. Public Release of the Dataset: We understand the importance of dataset accessibility for reproducibility and wider academic impact. We plan to release the annotated dataset upon acceptance of the paper for transparency and facilitate further research in this field.
>
> 6. Experimental Validation and Ablation Studies: Your point regarding the lack of ablation studies is well-taken. We will include an ablation study that dissects the contributions of individual components, such as the segmentation encoder and the CNN backbone, to provide empirical validation of their impact on performance.

---

### Official Review · Reviewer_dAEG · 2024-11-04

**Soundness:** 1
**Presentation:** 1
**Contribution:** 1
**Rating:** 3
**Confidence:** 5

**Summary:**

This paper presents "FLAIR", a foundational model designed for the grapheme recognition of the Indus Valley script, an ancient un-deciphered writing system. Recognizing the limited availability of labeled data, the authors leverage few-shot learning (FSL) through prototypical networks enhanced with a custom segmentation encoder called ProtoSegment.

**Strengths:**

The only positive aspect of this article is the topic.

**Weaknesses:**

(i) No details on IVC dataset.

(ii) What is the functionality of the protosegment model is also not properly illustrated  and hence the key contribution (if any at all  ) also cannot be perceived. This part should  have been aided with more illustrative diagrams.

(iii) Sloppy text - for example in page 3 line 147-148.

(iv) Extremely poor language and sentence formation.

(v) irrelevant references - just for the sake of filling up the paper , for example Line 44 in page 1.

**Questions:**

Seems the method described in this paper is just an off-the shelf algorithm - Could you specify what was the real contribution?

Where from did this IVC dataset was procured?

What was the reason for putting the details of the grant in the acknowledgement section??  This is completely against ICLR submission policy as this might reveal the authors identity.

**Details Of Ethics Concerns:**

The authors  have provided details on the grant which funded their research  in an acknowledgment section, this could reveal their identity.  Hence I believe  this is violating ICLR submission policy.

---

> ### Author Response · Authors · 2024-11-12
>
> Thank you for your detailed review and for pointing out critical areas of improvement. Your feedback is highly valuable in refining our manuscript. Below, we respond to your comments and outline the revisions we will implement:
>
> 1. Details on the IVC Dataset: We acknowledge that the current manuscript lacks sufficient detail about the IVC dataset. In response, we will provide a comprehensive description of the dataset's source, content, and annotation process. This will include details about how the dataset was curated from Parpola’s CISI volumes and Mahadevan’s seminal work, and how manual annotation was performed to ensure data quality.
>
> 2. Functionality and Illustration of the ProtoSegment Model: We understand that the description of the ProtoSegment model may not have been sufficiently detailed. To address this, we will include a more thorough explanation of the model’s functionality, emphasizing how the integration of a segmentation encoder contributes to enhanced feature extraction.
>
> 3. Sloppy Text and Language Quality: We appreciate your feedback on the language and writing quality. We will conduct a comprehensive revision of the manuscript to improve the clarity, coherence, and academic quality of the text. Specific instances, such as the noted section on page 3 (lines 147-148), will be rephrased for better readability​.
>
> 4. Relevance of References: We will carefully review all references to ensure they contribute meaningfully to the content.
>
> 5. Ethical Concerns Regarding Acknowledgment Section: Thank you for highlighting the potential ethical issue regarding the acknowledgment section. We will modify this section to comply with ICLR’s double-blind submission policy by removing or anonymizing any grant details to avoid revealing author identity.
>
> Responses to Specific Questions:
>
> Real Contribution of the Model: The key contribution of ProtoSegment lies in its unique integration of a segmentation encoder within the Prototypical Networks framework. This allows for more refined feature extraction and segmentation of input images, particularly benefiting tasks with limited data. We will expand the discussion on this novelty to clearly differentiate it from existing methods​.
>
> Procurement of the IVC Dataset: The IVC dataset was curated from sources such as Parpola’s CISI volumes and Mahadevan’s "The Indus Script: Texts, Concordance and Tables."

---

### Note · Authors · 2025-08-01

I have read and agree with the venue's withdrawal policy on behalf of myself and my co-authors.

---

### Meta-Review · Area_Chair_LGuB · 2024-12-20

**Metareview:**

The authors present ProtoSegment, a method based on prototypical networks for grapheme recognition in Indus script images from a curated Indus Valley Civilization (IVC) dataset. Their experimental results demonstrate that ProtoSegment achieves performance comparable to the state-of-the-art on few-shot classification tasks using the Omniglot dataset, and surpasses Prototypical Networks when tested on the IVC dataset.

While the research topic is undoubtedly interesting, there are significant concerns regarding both the novelty of the proposed method and the lack of detailed information about the curated dataset. ProtoSegment's architecture is fundamentally derived from Prototypical Networks, with only incremental modifications to the original approach. Furthermore, the authors provide insufficient details about the IVC dataset, making it impossible to evaluate whether this dataset constitutes a meaningful contribution to the field. Although the authors addressed some reviewers' comments during the rebuttal phase, their responses did not adequately resolve the core concern about the method's limited originality.

**Additional Comments On Reviewer Discussion:**

The reviewers raised several significant concerns about this paper, focusing on three main areas: the limited novelty of the proposed method, insufficient details about the curated dataset, and the overall poor writing quality and clarity. While the authors provided some responses during the rebuttal phase, they failed to adequately address any of the critical concerns raised by the reviewers. Furthermore, the authors did not revise their manuscript to incorporate the reviewers' feedback.

---

### Decision · Program_Chairs · 2025-01-22

Reject